# Serological analysis in humans in Malaysian Borneo suggests prior exposure to H5 avian influenza near migratory shorebird habitats

Hannah Klim [1] ✉, Timothy William[2,3,4], Jack Mellors[1], Caolann Brady[1], Giri S. Rajahram [4], Tock H. Chua [5,6], Helena Brazal Monzó[7], Jecelyn Leslie John[8], Kelly da Costa [9], Mohammad Saffree Jeffree [10], Nigel J. Temperton[9], Tom Tipton[1], Craig P. Thompson [11], Kamruddin Ahmed [8,12,13], Chris J. Drakeley [7], Miles W. Carroll [1] & Kimberly M. Fornace[7,14] ✉

Cases of H5 highly pathogenic avian influenzas (HPAI) are on the rise. Although mammalian spillover events are rare, H5N1 viruses have an estimated mortality rate in humans of 60%. No human cases of H5 infection have been reported in Malaysian Borneo, but HPAI has circulated in poultry and migratory avian species transiting through the region. Recent deforestation in coastal habitats in Malaysian Borneo may increase the proximity between humans and migratory birds. We hypothesise that higher rates of human-animal contact, caused by this habitat destruction, will increase the likelihood of potential zoonotic spillover events. In 2015, an environmentally stratified cross-sectional survey was conducted collecting geolocated questionnaire data in 10,100 individuals. A serological survey of these individuals reveals evidence of H5 neutralisation that persisted following depletion of seasonal H1/H3 HA binding antibodies from the plasma. The presence of these antibodies suggests that some individuals living near migratory sites may have been exposed to H5 HA. There is a spatial and environmental overlap between individuals displaying high H5 HA binding and the distribution of migratory birds. We have developed a novel surveillance approach including both spatial and serological data to detect potential spillover events, highlighting the urgent need to study cross-species pathogen transmission in migratory zones.

Cases of highly pathogenic avian influenza (HPAI) are on the rise in poultry and wild bird populations. Of particular concern is the influenza A(H5N1) clade 2.3.4.4b, which has been responsible for a striking increase in deaths in wild birds and domesticated poultry since 2020[1]. The World Organisation for Animal Health estimated that in 2022 at least 131 million domestic poultry succumbed to the virus or were culled due to exposure to the virus[2].

Migratory shorebirds and waterfowl are known hosts for avian influenza viruses (AIVs), including H5N1[3,4]. The global movements of these species can facilitate the dispersion and increased genetic diversity of the virus. Climate change and extreme weather events may be one cause of the recent increased pervasiveness of H5N1[4,5]. Habitat destruction resulting from these environmental changes, including coastline destruction, influences the distribution of migratory species, which in turn impacts the dynamics of viral spread[4-6]. As habitats and food sources become increasingly scarce, there is potential for higher contact rates amongst different migratory avian species and between wild and domesticated birds. This increased contact creates more

opportunities for viral reassortment and for the virus to jump between hosts.

In 2022 and 2023, several notable spillover events into mammals were caused by 2.3.4.4b H5N1 viruses[7], including seals[8], mink[9], and sea lions[10]. There is no evidence of sustained community transmission amongst humans, with the number of human cases remaining limited[2,11]. Mammal-to-mammal spread has been observed in the ongoing H5N1 dairy cow outbreak in the United States and in mink in Spain in 2022[12,13]. As human cases of H5N1 can be clinically severe with a high mortality rate, there are concerns that the virus will reassort or mutate to enable community transmission[1]. To prevent further viral spillover, it is crucial to understand current rates of H5N1 exposure. If there are subclinical or non-severe cases of H5N1, the symptoms of such cases may be indistinguishable from that of other influenza subtypes, so exposure to the virus would remain undetected unless an infected individual receives medical care. Therefore, the global prevalence of H5N1 exposure in humans is presently unknown and is likely to be underestimated[14]. Consequently, there is an urgent need to study serological exposure to these H5N1 viruses in humans.

We investigated H5N1 in humans in Malaysian Borneo, due to a unique confluence of environmental and ornithological factors in the region. Borneo is an important migratory stop for wild shorebirds transiting along the East-Asian Australasian flyway, which spans from Russia to Australia. Malaysia contains 13 important stopover sites for 20 species of migratory shorebirds, with four such sites in Malaysian Borneo[15]. Although Malaysian Borneo has an extensive coastline, it is also a region that has experienced much environmental change in the last 50 years[16,17]. Between 1973 and 2015, the island of Borneo lost an estimated 50% of its forests[16]. Habitats that would usually be home to these migratory shorebirds, including freshwater swamp forests[18], mangrove forests[19], and peat swamp forests[20], have been cleared and logged to make room for industrial agricultural efforts[15].

Several of the migratory species known to transit through Malaysian Borneo are carriers of various influenza A virus (IAV) subtypes, including H5N1 viruses[21–23]. However, the only recorded outbreak of HPAI H5N1 in Malaysian Borneo to date was a clade 2.3.2.1c virus outbreak in a commercial poultry farm in 2018 in the Sabah state[24]. An estimated 30,000 birds were either culled or succumbed to the virus during a month-long period[25]. Although this poultry outbreak was severe, there have been no human cases of H5N1 reported in Malaysian Borneo.

Here, we perform a serological survey of human influenza exposure in Sabah, Malaysian Borneo, to examine the immunological footprint of H5N1 in the region. We additionally present a method for minimising the impact of IAV hemagglutinin (HA) subtype cross-reactivity on serological results. We define species distributions of domesticated poultry and migratory wild shorebirds and demonstrate that environmental covariates can be used as a proxy to model wild shorebird contact. We additionally identify shared spatial distributions and environmental risk factors between the presence of migratory shorebirds and clade-specific H5N1 seroprevalence using a Bayesian framework. This study highlights the need to increase surveillance for rare zoonotic diseases at migratory sites and presents an approach for modelling the distributions of serological results and reservoir species.

## Results

### Identification of heterogeneous patterns of influenza binding and pseudo-neutralisation

In 2015, 10,100 individuals from 2650 households in Northern Sabah were sampled[26]. The surveyed population age ranged from 3 months to 105 years and was 47% male ($n = 4776$) and 53% female ($n = 5324$)[27]. The locations of the sampled villages and shorebird sightings throughout Sabah are displayed in Fig. 1. 14% ($n = 359$) of these households were situated within 10 km of previously recorded wild

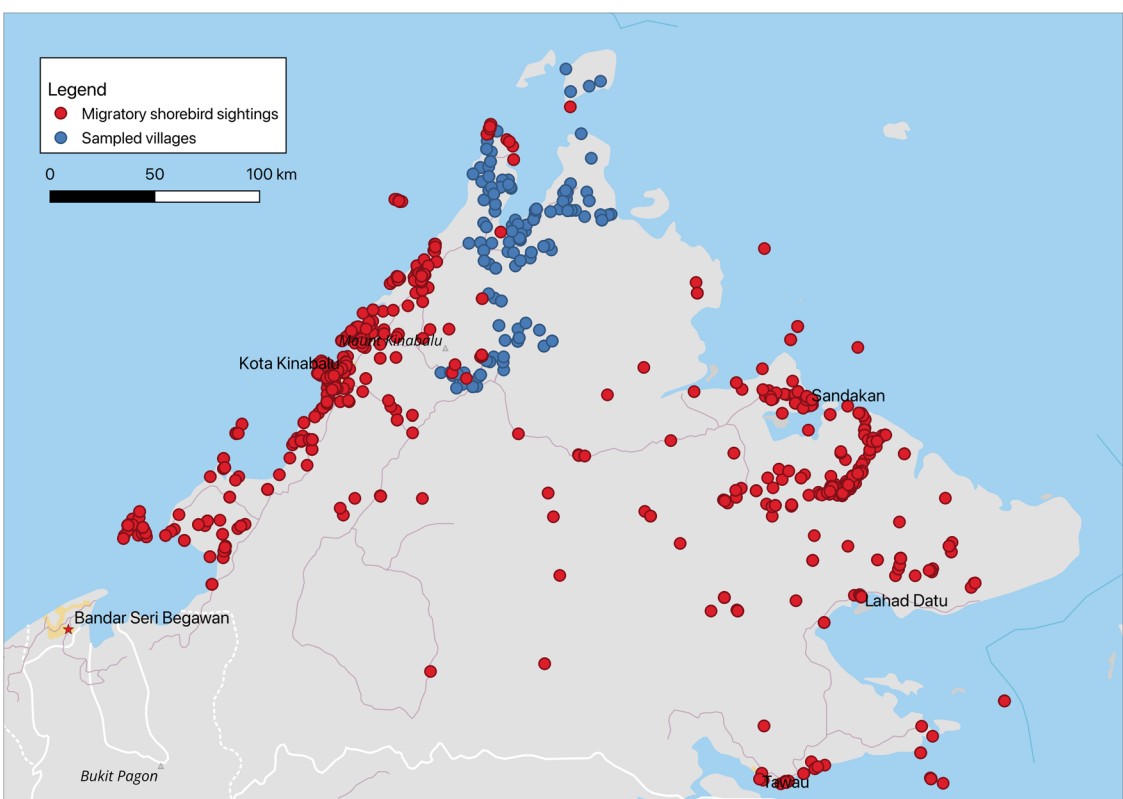

**Fig. 1 | Distribution of sampled villages (blue) and migratory shorebird sightings (red) in Sabah.** Sampled villages are in the Northern Sabah region, while migratory shorebird sightings are distributed throughout Sabah. This figure was generated in QGIS 3.30.2[62]. Natural Earth raster map data was used to generate the canvas map. Source data for migratory shorebird sightings are provided as a Source Data file.

shorebird sightings. 51% of the population are poultry owners ($n = 5137$), mostly evenly distributed throughout the entire sample region[27].

2000 of these individuals were randomly selected for serological testing of IAV exposure by enzyme-linked immunosorbent assays (ELISAs), which measure antibody (IgG) binding to HA. HA is a glyco-protein and the dominant immunogenicity in IAV. 500 individuals were selected from each of the four geographic regions sampled in the study: Kudat, Pitas, Ranau, and Kota Marudu (Supplemental Fig. 1). The mean age of the selected samples was 29. By region, the mean age in the selected samples from Kudat is 29, in Pitas the mean age was 27, in Ranau mean age was 27, and in Kota Marudu the mean age was 31. The study population was exposed to two known potential sources of avian influenza spillover: domesticated poultry and wild shorebirds. 23% ($n = 454$) of the randomly selected individuals lived within 10 km of a migratory shorebird sighting, and 51% ($n = 1024$) were poultry owners. 47% ($n = 217$) of individuals living within 10 km of migratory shorebird sightings were also poultry owners. The categories of poultry owners and those living near these migratory sites were used to examine differences in the binding and later, neutralisation of H5 viruses.

We used four HAs as our viral antigens for ELISAs. Antibody binding was reported in absorbance units (AU), where higher AU indicates increased antibody binding to the HA.

Two HAs were used in our ELISAs to capture binding to the sub-types of seasonal viruses that would have been circulating prior to sample collection in 2015: H1N1 (A/California/07/pdm2009) and H3N2 (A/New York/55/2004). Two H5 antigens were used to examine binding to avian viruses. A/Duck/Laos/3295/2005(H5N1) belongs to the H5 clade 2.3.4. This clade was circulating in Asia from 2003 to 2013 in avian species before breaking off into sub-lineages[28]. A/snow goose/Missouri/CC15-84A/2015(H5N2) belongs to clade 2.3.4.4, which did not become the predominant avian flu clade in Asia until 2018[29]. There are 38 amino acid differences between these two viral HAs (~93% identity) (Supplementary Table 1).

Seasonal influenza binding was evenly distributed throughout the population, irrespective of exposure to potential IAV reservoirs (Fig. 2a). Median H3N2 binding was 1.437($\pm$0.035, standard deviation) AU, and there were no differences when the population was categorised by poultry ownership or proximity to wild shorebirds. There were a wide variety of H3N2 responses, with 23 above 3 AU, 517 above 2 AU, and 1414 above 1 AU. H1N1 binding was not as strong in the study population (median of 0.3287($\pm$0.0109) AU). Poultry owners within the population had a slightly higher mean rank binding to the H1N1 HA than non-owners, 0.31 versus 0.29 AU ($p = 0.0174$) by a two-tailed Mann–Whitney test. Two responses were above 2AU and 22 were above 1AU.

There were no significant differences ($p < 0.05$) in H5 binding to either HA when the population was sorted by poultry ownership. However, those living within 10 km of wild shorebirds had a statistically significantly higher mean rank than those living further away for both H5 antigens by a Mann–Whitney test. Population medians were 0.25 and 0.21 AU ($p = 0.0012$) for the 2.3.4 pre-2015 virus and 0.12 and 0.11 AU ($p = 0.0156$) for the post-2015 2.3.4.4 virus. Responses to A/snow goose/Missouri/CC15-84A/2015(H5N2) were modest, with one response above 2 AU and only 10 above 1 AU. A/Duck/Laos/3295/2005(H5N1) was a more reactive antigen with one response above 3 AU, 20 above 2 AU, and 161 responses above 1 AU. High binding to one antigen did not correlate with similar binding to a different antigen. The Spearman's rank correlation coefficient ($r$) of binding to H5 antigens was $r = 0.22$, and the other correlations ranged from $r = -0.03$ to 0.13 (Supplemental Fig. 2a). The Scottish blood donor cohort showed higher correlations between the H5 and the seasonal H1 antigens ($r = 0.19$, $r = 0.25$) and almost no correlation between the H5 antigens ($r = 0.02$) (Supplemental Fig. 2b). These Spearman $r$ values indicate that the binding to one ELISA antigen was not linearly related

to binding of a different antigen. These results also suggest that there are population-level differences in HA binding between the Scottish and Malaysian cohorts.

204 samples were selected for follow-up pseudotyped micro-neutralisation assays (PNA) against two viruses from prior to 2015: A/Indonesia/05/2005(H5N1) and A/Bar headed goose/Qinghai/1 A/2005(H5N1). The generated pseudotyped lentiviruses displayed HA on their surface but were replication-deficient[30]. This allowed for the study of the ability of the cohort plasma to inhibit viral growth by binding to HA without the risk of working with a fully replicating virus.

The 20 samples that exhibited binding above 2 AU to A/Duck/Laos/3295/2005(H5N1) were chosen for inclusion in subsequent neutralisation experiments, and the remaining samples were randomly selected based on the quantity of available pseudovirus. 52% ($n = 108$) of the population tested for neutralisation were poultry owners, and 34% ($n = 69$) lived within 10 km of a shorebird sighting. The neutralising abilities of a sample were evaluated based on the NT50 value, which is the 50% inhibitory concentration (i.e. the concentration of serum or plasma necessary to achieve 50% virus inhibition). Previous studies have used 1:100 as a cut-off for seropositivity[30], but instead the highest NT50 from a cohort of negative control samples was used here to reduce the likelihood of including cross-reactive neutralising (false positive) samples in subsequent experiments (1:1173 for A/Indonesia/05/2005 and 1:1015 for A/Bar headed goose/Qinghai/1A/2005), as in Thompson et al.[31] (Supplemental Fig. 3).

Three samples were considered seropositive in their neutralising ability (1.5% seropositivity rate) against A/Indonesia/05/2005(H5N1) pseudovirus, of which two (67%) were poultry owners and all three (100%) lived within 10 km of wild shorebird migratory sites (Fig. 2b). The samples were 6.8% seropositive ($n = 14$) against A/Bar headed goose/Qinghai/1A/2005(H5N1) pseudovirus. Six of these seropositive individuals owned poultry, and all 14 lived in proximity to wild shorebirds. When comparing differences in NT50, there were no significant differences based on poultry ownership to either virus. Individuals with proximity to wild shorebirds did have a statistically significant increase in mean rank NT50 over those living further away, 210.4 versus 59.24 ($p < 0.0001$) by a Mann–Whitney test.

We then sought to determine whether the individuals in the rural Malaysian cohort had serological responses to the virus responsible for the 2018 outbreak (Fig. 2b). Another H5N1 pseudovirus presenting A/chicken/Malaysia (Sabah)/6123/2018 HA (Clade 2.3.2.1c) was generated for these experiments. Of the 204 samples originally tested for neutralisation only 178 had sufficient volume to be repeated with two replicates in this assay. 50% of the 178 samples were poultry owners ($n = 89$) and 33% ($n = 58$) lived within 10 km of a migratory shorebird. 12 samples neutralised the pseudovirus above the seropositivity cut-off (NT50 > 1:1358) (Supplemental Fig. 3c). This is a seropositivity rate of 6.7%, which is akin to the observed rate for A/Bar headed goose/Qinghai/1 A/2005(H5N1) pseudovirus of 6.8%. 50% ($n = 6$) of the seropositive samples against A/chicken/Malaysia (Sabah)/6123/2018(H5N1) pseudovirus were individuals living within 10 km of a wild shorebird migratory site. 75% ($n = 9$) of the seropositive samples were from poultry owners.

## Persistence of pseudo-neutralisation upon depletion of cross-reactive antibodies

We developed a bead-based antibody depletion assay to determine the role of cross-reactivity in these neutralising responses. We conjugated magnetic non-fluorescent beads to several HA for seasonal influenza strains (Supplemental Fig. 4). The beads were then incubated with the plasma. Using a magnet to pull down seasonal influenza antibody-bound beads, unbound antibodies can be extracted. This plasma extract is depleted of cross-reactive seasonal influenza responses and was tested in neutralisation assays for a maintained H5 response.

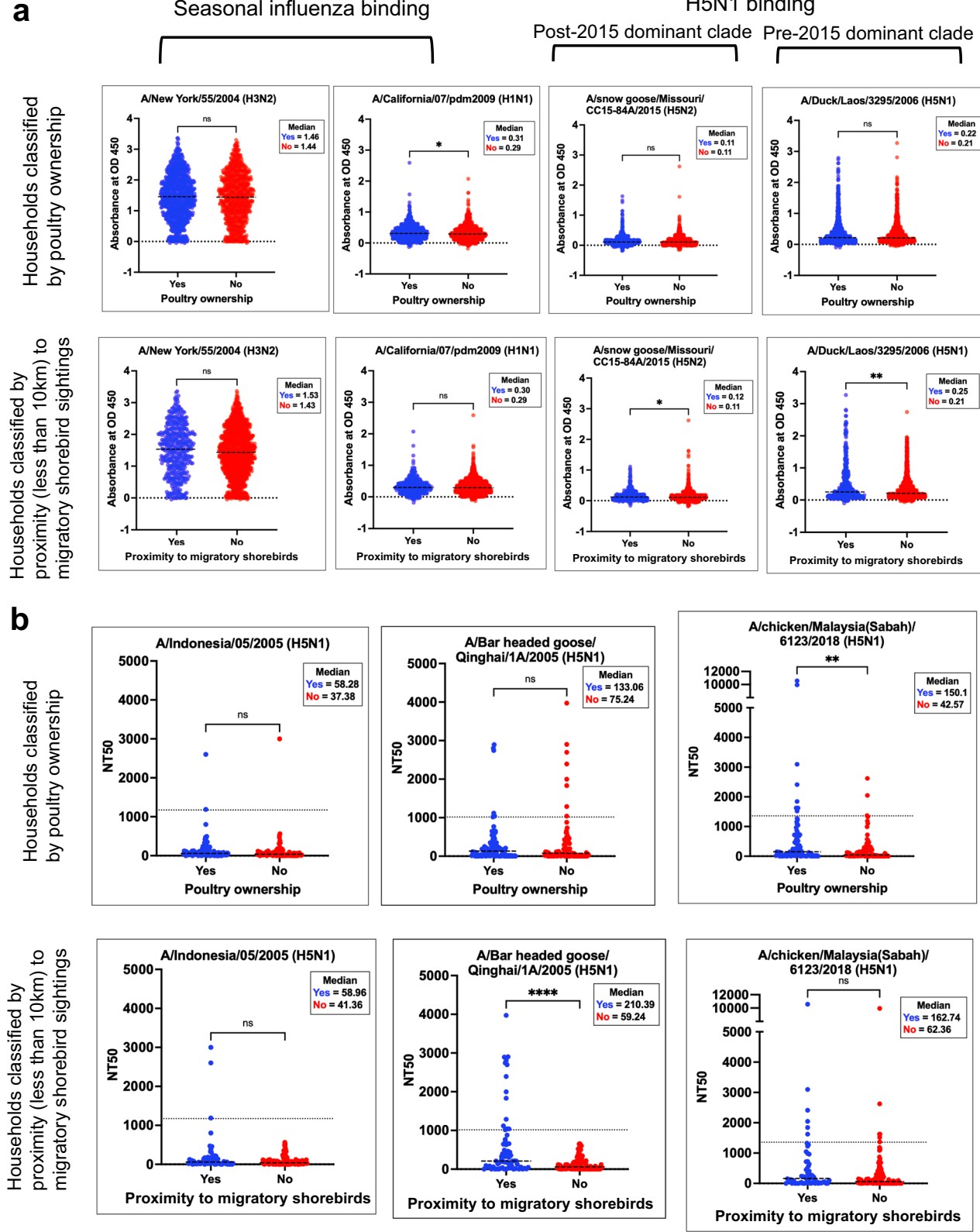

Neutralisation assays were repeated with the treated plasma against A/Bar headed goose/Qinghai/1A/2005(H5N1).

Pooled positive control serum was obtained from a vaccination trial (vaccinated with A/Indonesia/05/2005(H5N1)) from BEI Resources (NIH, NIAID). The two negative control samples were low and high titre human H1N1 convalescent sera and were assumed not to have been exposed (i.e. the observed neutralisation is likely due to cross-reactive antibodies). The high and low tire responses are relative to each other according to the supplier (BEI Resources). Neither positive pool experienced a statistically significant change in NT50 before and after treatment (plasma dilutions of 1:1915 vs. 1:1857 and 1:2436 vs. 1:2166) (Fig. 3a). The first negative sample (low titre) experienced a 72% reduction in NT50 after treatment (1:595.1 reduced to 1:163.7), which was considered statistically significant ($p = 0.0010$) via a sum-of-

**Fig. 2 | Individuals living in proximity to migratory shorebirds have increased H5 HA binding and neutralisation.** As the data is non-parametric, the medians of each group were compared using a two-tailed Mann–Whitney test in GraphPad Prism 10.0.3. **a** Represents ELISA-binding data of $n = 2000$ samples, where the population is 51% ($n = 1024$) poultry owners and 23% ($n = 454$) were living in proximity to wild shorebirds (<10 km). Statistically significant *p*-values are 0.0174 (H1N1 binding, poultry ownership), 0.0156 (H5N2 binding, proximity), and 0.0012 (H5N1 binding, proximity). The nonsignificant *p*-values were 0.4707 (H3N2 binding, poultry ownership), 0.0612 (H3N2 binding, proximity), 0.7503 (H1N1 binding, proximity), 0.7633 (H5N2 binding, poultry ownership), and 0.1997 (H5N1 binding, poultry ownership). All samples were tested in duplicate at a 1:50 dilution. **b** Displays NT50 values for H5N1 pseudovirus microneutralisation assays ($n = 204$ for A/Indonesia/05/2005 and A/Bar headed goose Qinghai/1A/2005; $n = 174$ for A/

chicken/Malaysia (Sabah)/6123/2018). NT50 is here defined as the dilution factor of the serum necessary to reach 50% inhibition of the pseudovirus. The samples tested against A/Indonesia/05/2005 and A/Bar headed goose Qinghai/1A/2005 pseudo-viruses were 52% ($n = 108$) poultry owners, and 34% ($n = 69$) lived within 10 km of a shorebird sighting. The samples tested against A/chicken/Malaysia (Sabah)/6123/2018 pseudovirus ($n = 178$) were 50% poultry owners ($n = 89$), and 33% ($n = 58$) lived within 10 km of a migratory shorebird. The *p*-value for proximity comparisons for H5N1 Qinghai NT50s was <0.0001 (0.000007), and the *p*-value for poultry ownership for H5N1 Sabah NT50s was $p = 0.0035$. The nonsignificant p-values were 0.0950 (H5N1 Indonesia, poultry ownership), 0.1461 (H5N1 Indonesia, proximity), 0.4328 (H5N1 Qinghai, poultry ownership), and 0.0687 (H5N1 Sabah, proximity). The figure was generated in GraphPad Prism 10.0.3. Source data are provided as a Source Data file.

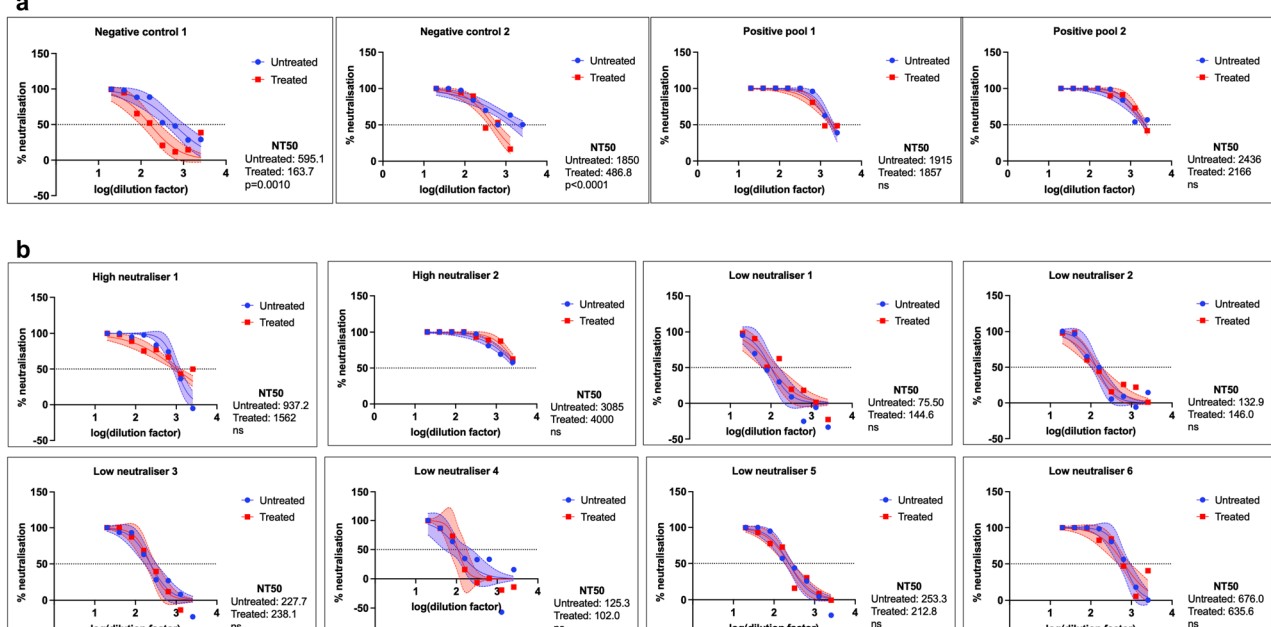

**Fig. 3 | H5 neutralisation is not depleted upon treatment with seasonal flu beads.** NT50 values were calculated using nonlinear regression in GraphPad Prism 10.0.3. Each plot displays the nonlinear curve fit used to calculate NT50 and 95% confidence interval bands of the nonlinear curve fit. Each plot is the average of two replicates minimum (Negative 1, $n = 3$; Negative 2, $n = 5$). LogNT50 values before and after treatment were compared via a sum-of-squares *F* Test in GraphPad (two-sided). All *p*-values for statistically significant differences in logNT50 are listed, while non-significant ($p > 0.05$) changes are listed as ns. **a** Displays the control serum for the depletion assay. Negative controls 1 and 2 are H1N1 convalescent sera (low and medium titre, respectively), while positive control pooled serum was exposed to a sub-virion H5N1 vaccine. Both sets of controls were obtained through

BEI Resources. Positive pool 1 is a medium titre pool and positive pool 2 is a low titre pool, according to the supplier. For negative controls 1 and 2, the change in NT50 was statistically significant ($p = 0.0010$ and 0.00004, respectively), while for the positive pools, these changes were not significant ($p = 0.8467$ and 0.5636, respectively). Eight samples with unknown exposure history from the Malaysian cohort were treated by the same method in **b**. All samples were treated with five rounds of fresh protein-conjugated beads in a cocktail mixture at a concentration of 100 beads/µL per bead and aspirated upon incubation with a magnet. The changes in NT50 were not significant for any of these samples ($p = 0.0958$, 0.3534, 0.0813, 0.7387, 0.8763, 0.6636, 0.4321, 0.8414 for high neutraliser 1, 2, and low neutraliser 1 through 6, respectively). Source data are provided as a Source Data file.

squares F Test comparing the logNT50 values. The NT50 for negative control 2 (high titre) experienced a 74% reduction upon treatment with the beads, from 1:1850 to 1:486.8, which was also statistically significant ($p < 0.0001$).

Only two of the fourteen original high neutralisers (NT50 > 1000) from our sample cohort could be tested via this method (due to limited plasma volume), along with six low neutralisers where the NT50 was above 100 but below 1000. Neither the high nor low neutralisers experienced a statistically significant reduction in NT50 after treatment with the beads (Fig. 3b). This suggests that the neutralising response measured in these Malaysian Borneo samples may be H5-specific and not a cross-reactive neutralising response stemming from seasonal influenza exposure.

## Spatial and environmental overlap of H5 binding and wild bird distributions

From July 1979 to August 2023, there were 86,502 observations of the 20 species of shorebirds known to migrate through Malaysian Borneo in the Sabah region[15,32]. These observations were submitted to eBird from 7636 unique entries. The cumulative abundance of each species ranged from 2 to 28,631 observations (Table 1). The distributions of shorebirds were included across the whole-time frame of data availability (1979–2023), rather than just the year of sample collection (2015), to model species distribution and reduce year-to-year variability.

There are regions of high and low domesticated poultry ownership distributed throughout the Northern Sabah region, while wild

**Table 1 | Abundance of species observed in the Sabah region of Malaysian Borneo from July 1979 to August 2023 and reported to the eBird database[32]**

| Common name | Scientific name | Observations by birdwatchers |
|---|---|---|
| Asian Dowitcher | *Limnodromus semipalmatus* | 45 |
| Black-tailed Godwit | *Limosa limosa* | 284 |
| Broad-billed Sandpiper | *Calidris falcinellus* | 110 |
| Common Greenshank | *Tringa nebularia* | 2377 |
| Common Redshank | *Tringa totanus* | 2361 |
| Common Sandpiper | *Actitis hypoleucos* | 17,144 |
| Curlew Sandpiper | *Calidris ferruginea* | 421 |
| Eurasian Curlew | *Numenius arquata* | 187 |
| Far Eastern Curlew | *Numenius madagascariensis* | 195 |
| Greater Sand-Plover | *Charadrius leschenaultii* | 2976 |
| Lesser Sand-Plover | *Charadrius mongolus* | 3851 |
| Long-toed Stint | *Calidris subminuta* | 10,791 |
| Marsh Sandpiper | *Tringa stagnatilis* | 4739 |
| Pacific Golden-Plover | *Pluvialis fulva* | 6223 |
| Red-necked Stint | *Calidris ruficollis* | 3613 |
| Spotted Redshank | *Tringa erythropus* | 2 |
| Temminck's Stint | *Calidris temminckii* | 141 |
| Terek Sandpiper | *Xenus cinereus* | 2114 |
| Whimbrel | *Numenius phaeopus* | 2097 |
| Wood Sandpiper | *Tringa glareola* | 26,831 |

shorebird contact and high H5 binding are confined to several distinct areas (Fig. 4a, Supplemental Fig. 6). There is a clear section of overlap between wild shorebird contact and high H5N1 binding in the north-western portion of our sample region, in a district known as Kudat. This high H5N1 binding cohort included 20 samples that have an AU > 2. An AU > 2 was considered high binding as the highest response in the control cohorts had AUs of 1.28 (urban Malaysia cohort) and 0.47 (Scottish control cohort) (Supplemental Fig. 5). ELISA binding was chosen for this spatial analysis over neutralisation, as this allowed us to include data on binding at 2000 locations.

Distributions of wild shorebird contact and H5N1 binding (by ELISA to A/Duck/Laos/3295/2006) were then plotted as probabilities of species distribution or antibody binding (Fig. 4a). We identified environmental risk factors for shorebird distribution and ELISA binding, reported as adjusted odds ratios based on generalised linear mixed effects modelling (Fig. 4b, Supplemental Figure 8). The model of H5 binding (Supplemental Table 2) and of the wild shorebird distribution (Supplemental Table 3) had several shared environmental risk factors (Fig. 4b). Closeness to the sea increases the odds of wild shorebird contact by nearly 10 times, and the odds of high H5 binding over 800 times. Both H5 binding and wild shorebird sightings were more likely in areas with lower elevation and lower precipitation. Proximity to forested areas increased the odds of wild bird sightings along with Euclidean distance from roads (a measure of remoteness). Increased distances from irrigated farmland also increased the odds of H5 binding. The risk factors which associated with H5 binding and wild shorebird contact are indicative of remote habitats and low populous areas. Inclusion of poultry ownership within the household or surrounding villages did not improve the model fit for H5 binding (Supplemental Table 4).

Using these environmental risk factors, we fit Bayesian models for wild shorebird species distributions and H5 ELISA data. All models had statistically significant spatial autocorrelation detected by Moran's *I* after the inclusion of fixed environmental effects. The wild shorebird distribution indicates clustered data, with a positive Moran's *I* value of

0.0191 (expected value of −0.0001, $p < 2.2e-16$). Similar results were observed for H5 binding, in which $I_{observed} = 0.1235$, $I_{expected} = -0.0005$, and $p < 2.2e-16$. Based on Moran's *I*, we then fit geostatistical models, including spatially structured random effects. H1 ($p = 0.6692$), H3 ($p \approx 1$), and H5N2 ($p = 0.1641$) binding did not show spatial auto-correlation via Moran's *I*, so the results of these assays were not included in further geospatial analysis.

We subsequently fit joint spatial models of wild shorebird distribution and H5 binding. Incorporating a shared spatial random effect between shorebird distribution and H5 binding improved model performance, suggesting common spatial processes between H5 exposure and migratory shorebird distribution (Supplemental Tables 5 and 6).

## Discussion

Our results show evidence of heterogenous serological responses to avian influenza in Sabah. These results are spatially correlated and follow the distribution and habitats of migratory wild shorebirds over domesticated poultry in the region. As contact with avian species is currently necessary for a spillover event, it is critical to consider these migratory sights as key interfaces in stopping the viral spread.

Infection or vaccination with one influenza subtype is thought to generate memory B cells which could produce antibodies that bind to shared epitopes on other HAs[33,34]. The extent and pervasiveness to which these so-called cross-reactive antibodies exist across multiple HA subtypes and virus strains is not fully understood[35]. Seasonal IAV vaccination in Malaysia is limited, which removes one possible source of H5 cross-reactive antibody development[36]. Azami et al. reported 5.5% and 26.3% seasonal vaccine coverage in the elderly and healthcare workers in Malaysia[37]. Comparatively, in Scotland, the seasonal IAV vaccine coverage is 75.7% in the elderly and 42.4% in healthcare workers[38]. Although national vaccination rates are higher in Scotland, the Scottish blood donor cohort used in this study as a control cohort exhibited both lower antibody binding to H5 HAs and lower neutralisation titres against H5 pseudovirus (Supplemental Figs. 3, 5). This suggests that seasonal vaccination is unlikely to explain the high H5 responses in the rural Malaysian cohort.

In this study, there was a lack of correlations in the ELISA binding to different antigens (Fig. 2a and Supplemental Fig. 2). This may serve as evidence that the observed responses are not driven by the presence of cross-reactive antibodies. A study by Nachbagauer et al. showed that cross-reactive antibodies against H5 HAs were present after H1 infection in mice and that these antibodies could be detected by ELISAs[35]. The cross-reactive binding responses shown in Nachbagauer et al. to H5 after H3 infection were limited when compared to the H1 results, as H1 and H5 belong to the same HA phylogenetic group[34,39]. Antibody binding in the Scottish blood donor cohort, which is presumed to be H5 negative, follows the trend observed by Nachbagauer et al. (Supplemental Fig. 2). In the Scottish blood donor responses, the observed correlations between H1 and H5 HA binding ($r = 0.19$, $r = 0.25$) are greater than in the Malaysian cohort, where they were nearly zero ($r = 0.04$, $-0.03$). Conversely, the H5 HA binding in the Malaysian cohort has higher correlations with H3 HA binding ($r = 0.13$, $r = 0.13$) than the Scottish cohort ($r = -0.02$, $r = -0.06$). The binding results for the Malaysian cohort present an unusual finding, which deviates from the Scottish cohort and the expectations established in the literature. If the H5 HA binding responses were the result of cross-reactive antibodies caused by H1 HA exposure, we would expect a stronger positive correlation (similar to that of the Scottish cohort) and spatial relationship between the H1 and H5 HA binding results in the Malaysian cohort. The highest correlation in the Malaysian cohort study is between the H5 HA antigens. As these are from the same HA subtype, some level of H5 cross-reactivity is likely[33–35]. When compared with the results of the Scottish cohort, the evidence supports our

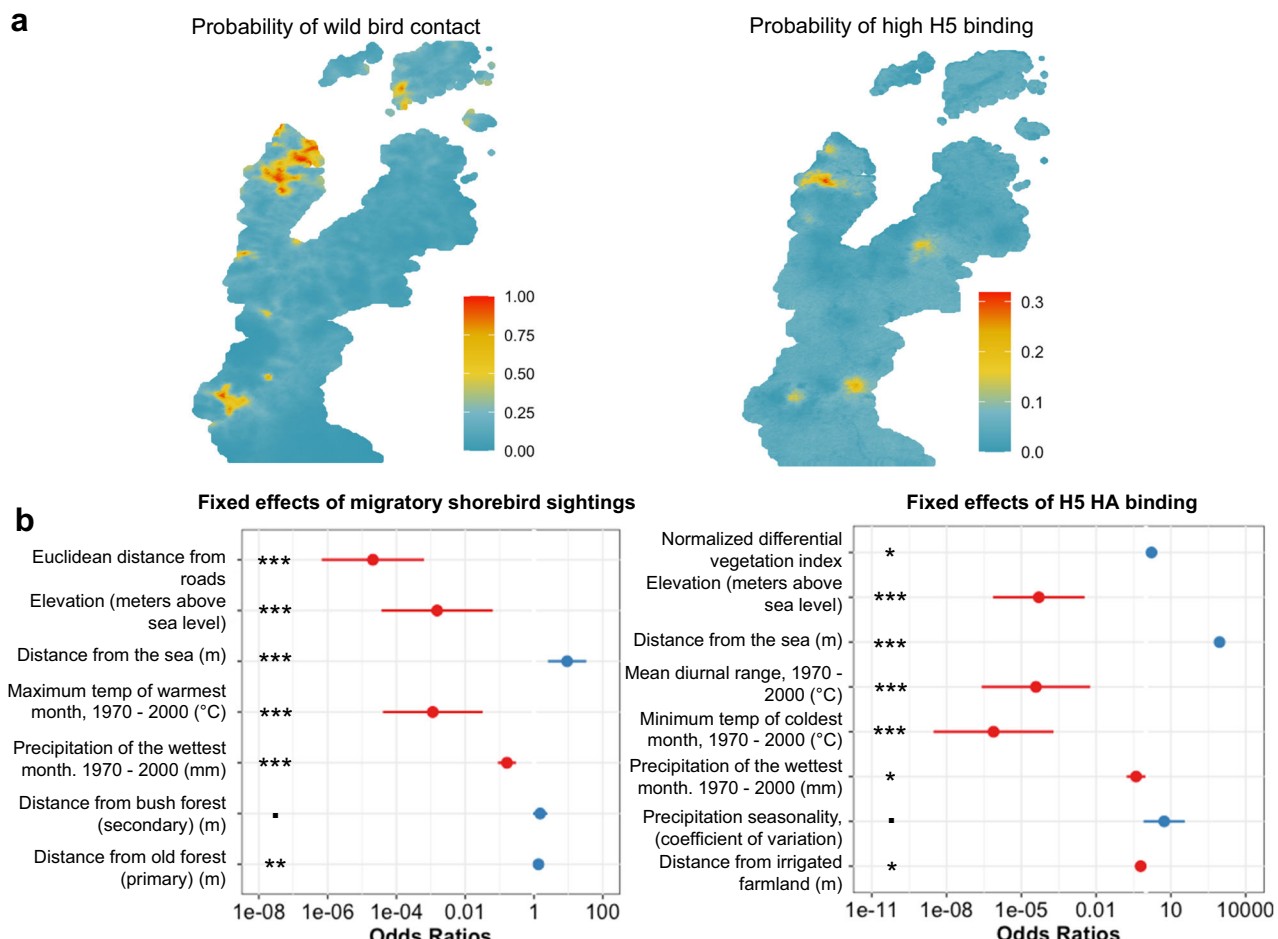

**Fig. 4 | Spatial overlap between wild shorebird contact and H5 binding exists that is not seen when comparing domesticated poultry contact and H5 binding. a** Estimated mean probability of wild shorebird contact and estimated mean probability of high H5 binding (to A/Duck/Laos/3295/2006) from Bayesian geostatistical models predicted across the study site in Northern Sabah (Fig. 1). Posterior probabilities were estimated using 1000 posterior samples. These posterior probabilities were then used to predict the probability of the outcome variable (H5 binding or species distribution) across the whole study region at 30 m spatial resolution. A total of $n = 2000$ samples were tested in duplicate by ELISA at a 1:50 dilution, and IgG binding to H5 HA was measured in absorbance units (AU). High H5 binding is considered greater than 2.0 AU based on comparison to the control cohorts. **b** Adjusted odds ratios for environmental predictors in mixed effects modelling of contact and H5 binding are shown along with 95% confidence intervals for each odds ratio and statistical significance, where $p \le 0.10$ (.), $p \le 0.05$ (*), $p \le 0.01$ (**), $p \le 0.001$ (***). Odds ratios greater than 1 (shown in blue) indicate a

factor that increases the odds of high H5 binding or of migratory shorebird sightings. Odds ratios <1 (shown in red) decrease the odds of those outcomes. *p*-values for the risk factors for the odds of wild bird sightings are $3.07 \times 10^{-6}$ (model intercept), $6.88 \times 10^{-10}$ (Euclidean distance from roads), 0.001 (elevation in meters above sea level), 0.001 (distance from the sea in meters), $6.90 \times 10^{-5}$ (maximum temperature of warmest month in °C), $4.68 \times 10^{-9}$ (precipitation of the wettest month in mm), 0.094 (distance from bush forest in meters) and 0.01 (distance from old forest in meters). *p*-values for the risk factors for the odds of H5N1 binding are as follows: 0.018 (normalised differential vegetation index), $3.93 \times 10^{-6}$ (elevation in meters above sea level), $1.67 \times 10^{-6}$ (distance from the sea in meters), $4.79 \times 10^{-5}$ (mean diurnal range in °C), $4.85 \times 10^{-7}$ (minimum temperature of coldest month in °C), 0.04 (precipitation of the wettest month in mm), 0.084 (precipitation seasonality), and 0.027 (distance from irrigated farmland in metres). Source data are provided as a Source Data file including mean posterior estimates, odds ratios, and confidence intervals.

hypothesis that the binding to H5 HA is not due to some general influenza reactivity or recent seasonal infection/vaccination and could be H5-specific.

H5 HA phylogeny adds further credibility to these binding results. A/Duck/Laos/3295/2005(H5N1) and 2.3.4 clade viruses were circulating in Asia for at least 10 years prior to sample collection[28]. If there was exposure to an H5 virus in the rural Malaysian study cohort, it is possible that it could have been a 2.3.4-like virus due to the timings of circulation. While a range of binding responses to the HA from 2.3.4 was observed, there were hardly any responses above AU 1 ($n = 11$, 0.55%) to the A/snow goose/Missouri/CC15-84A/2015(H5N2) 2.3.4.4 virus (Fig. 2a). As this second viral HA clade did not become dominant until 3 years after our sample collection[29], it is highly unlikely that members of the study cohort could have had an opportunity to be exposed to a 2.3.4.4 virus. The phylogenetic specificity of our binding results amongst H5

responses would not be observed if these responses were exclusively cross-reactive.

The observed trends in neutralisation are also supported by phylogenetic evidence. The 2.3.4 H5N1 ELISA antigen has more phylogenetic similarity with A/Bar headed goose/Qinghai/1A/2005(H5N1) (Clade 2.2, 18 amino acid differences) than A/Indonesia/05/2005(H5N1) (Clade 2.1.3.2, 20 amino acid differences) and A/chicken/Malaysia(Sabah)/6123/2018 (Clade 2.3.2.1c, 36 amino acid differences) (Supplementary Table 1). None of the high binding samples to Clade 2.3.4 HA by ELISA neutralised A/Indonesia/05/2005, one neutralised A/chicken/Malaysia (Sabah)/6123/2018, while 20% had NT50 > 1000 ($n = 4$) and 16 (80%) had NT50 > 100 against A/Bar headed goose/Qinghai/1A/2005. This suggests that the serological responses against H5 observed in this cohort are not general H5 responses. In fact, the responses are temporally and phylogenetically specific, which

suggests that the individuals in our cohort may have encountered a virus closely related to the H5 clades 2.3.4 or 2.2. Although the recombinant HAs used by ELISA are not matching strains to the pseudoneutralisation HAs, the observed binding and neutralisation are in accordance with the phylogenetic distance.

The persistence of these neutralising responses upon treatment with the seasonal flu-coated beads illustrates that the observed responses in the sample cohort were most similar to those of the H5 exposed positive controls (Figs. 2b and 3). Our study is limited in that a lack of PCR and health data makes it impossible to know if this population had genuine symptomatic cases of H5 avian influenza in this population. However, attributing the results of this study to previous seasonal influenza exposure would not fully explain the unusual responses we have observed. Previous work has highlighted the strong correlations between pseudovirus neutralisation and live virus neutralisation results for H5N1 IAV[30,40], further validating these results even without access to health data. We propose that our depletion assay could be useful to other researchers seeking to contextualise the role of cross-reactivity in their influenza serology results when clinical data is not available.

The geospatial distributions of the serological responses adds further support to the hypothesis of H5 exposure. Binding to seasonal influenza antigens exhibits a wide range of responses regardless of exposure to reservoirs, and this data is not spatially correlated (Fig. 2a). As seasonal influenza viruses are endemic, spatial homogeneity was expected. Indeed, if the seasonal influenza binding shared spatial trends with that of the avian influenza binding responses, it would be far more likely that the H5 results would be the result of cross-reactive antibodies from a previous seasonal influenza infection.

There was no spatial correlation in H5 2.3.4.4 binding. As the samples for this study were collected in 2015, this population would likely not have had the chance to be exposed to this post-2015 virus. Therefore, spatially correlated responses would have been highly unexpected. Instead, H5 2.3.4 binding was higher amongst those living near migratory shorebird locations, and the data are highly spatially correlated (Figs. 2a and 4a). These spatial patterns suggest that the H5 2.3.4 results are distinct from the other ELISA responses and may represent a footprint of some genuine previous exposure.

There are shared environmental risk factors between the wild shorebird distributions and H5 binding (Fig. 4b). The identification of proximity to the sea, low elevation (i.e. sea level) areas, closeness to the forest, and remote areas as risk factors indicates that wild migratory shorebirds were reported in their natural habitats[18–20], and that these habitats are also areas where binding to the H5 HA 2.3.4 antigen was highest. Shared environmental risk factors and the overlapping spatial distributions between wild shorebirds and H5 HA binding, suggest that some form of contact may be occurring between humans and wild shorebirds in these locations, whether direct or indirect. Future studies could collect contemporaneous data on wild shorebird movements and pathogen presence.

One mechanism of H5 spillover is the spread of the virus from wild shorebirds and waterfowl to domesticated poultry or swine and then into humans[41]. However, H5 2.3.4 binding is not related to poultry or swine contact in this study, as the inclusion of poultry or swine ownership as fixed effects did not improve model fit (Supplemental Table 7). This presents an unexpected finding, as poultry and swine farmers are often considered at risk of encountering IAV[41] (Supplemental Fig. 6). Additionally, there were no recorded waterfowl sightings in the Kudat district of Sabah, where several of the individuals with high antibody binding to H5 HA 2.3.4 reside, but shorebirds were recorded in this area (Supplemental Fig. 7). While there are many possible routes of influenza spillover, our findings highlight that proximity with wild migratory shorebirds could be an overlooked source of H5 transmission. The threat of spillover from migratory shorebirds was highlighted by Wille et al., who identified that red-necked stints (*Calidris ruficolis*), known to stopover in East and Southeast Asia on their way to Australia, had HA inhibitory antibodies to HPAI H5 viruses[23]. Although migratory shorebirds have traditionally been implicated in the spread of low-pathogenicity avian influenza viruses to poultry, Wille et al. highlight the possibility of migratory shorebirds' role in HPAI spread. Our study presents serological data which suggests that migratory shorebirds could already be playing a role in H5 spillover into humans[23,42]. Thees findings suggest a need to consider individuals living close to these migratory sites as a part of regular surveillance efforts to determine the risk of spillover in these locations.

Migratory shorebirds transiting between countries may carry influenza and expose individuals in the surrounding areas to avian viruses. These results presented in this study suggest that individuals living within 10 km of known migratory locations may have had previously unknown exposure to avian influenza of the H5 HA 2.3.4 or similar clade. As shorebird habits are being destroyed due to rising sea levels and land use changes, there is an urgent need to consider how this may force zoonotic reservoirs into closer contact with humans and increase the risk of HPAI spillover.

## Methods

### Samples collection and ethical approval

**Scottish blood donors.** A total of 63 serum samples from blood donors aged 17–80 collected by the Scottish National Blood Transfusion Service (SNBTS) in 2020 were used for this study. Ethical approval was obtained for the SNBTS anonymous archive−IRAS project number 18005. SNBTS blood donors gave fully informed consent to virological testing; donation was made under the SNBTS Blood Establishment Authorisation, and the study was approved by the SNBTS Research and Sample Governance Committee.

**Cross-sectional survey of rural Malaysian Borneo (Sabah).** Samples were collected, as a part of Fornace et al. [43] between September 17, 2015 and December 12, 2015, from individuals in the Pitas, Kudat, Ranau, and Kota Marudu districts of the Sabah region in Malaysian Borneo. All individuals in households selected for the environmental survey responses were asked to donate blood unless they were younger than 3 months old or could not be reached after three attempts. Whole blood was collected into precoated EDTA tubes (Becton-Dickinson, Franklin Lakes, USA). Samples were tested by ELISA and pseudoneutralisation assays after processing.

Written informed consent was obtained from all study participants or a legal guardian. The Medical Research Sub-Committee of the Malaysian Ministry of Health (NMRR-14-713-21117), the Research Ethics Committee of the London School of Hygiene & Tropical Medicine (8340), and the Oxford Tropical Research Ethics Committee (560-22) have approved this study.

**Urban Malaysian samples (Kota Kinabalu).** 678 samples were collected from blood donors in Kota Kinabalu, Malaysian Borneo from 2017 to 2020 for a cross-sectional study on Leptospirosis run in 2017 by Jeffree et al. [44]. These samples were tested by ELISA in Kota Kinabalu, Malaysia at the Borneo Medical Health Research Centre. Ethical clearance was obtained from the Ethics Committee of the Faculty of Medicine and Health Sciences, Universiti Malaysia Sabah, Kota Kinabalu, Sabah, Malaysia (UMS).

### Sample selection

The 2000 samples used for ELISAs in this study were chosen at random from the 10,100 samples in the rural cohort, including 500 samples each from the four districts in the study (Pitas, Kudat, Ranau, and Kota Marudu). The number of samples used at various points throughout this study is indicated by *n*.

## Viral proteins, antibodies, and antiserum

All viral proteins were obtained through BEI Resources, NIAD, NIH from various contributors (Supplemental Table 7). All HAs were produced in Sf9 insect cells using a baculovirus expression system with a variety of purification methods and tags. All antibodies and antiserum for IAV HAs were sourced externally (Supplemental Table 8).

## Detection of IgG binding by ELISA

The binding of antibodies to a variety of seasonal and avian haemagglutinin was measured by IgG ELISAs in a protocol adapted from Thom et al. [45].

MaxiSORP™ NUNC-Immuno plates were coated with 0.125 µg/mL of A/Duck/Laos/3295/2006(H5N1) (H5 Clade 2.3.4), A/snow goose/Missouri/CC15-84A/2015 (H5N1) (H5 Clade 2.3.4.4), and A/New York/55/2004(H3N2) or 1 µg/mL for A/California/07/pdm2009(H1N1) in 50 µL 1X PBS. Plates were incubated overnight at 4 °C. The following day, plates were blocked with 200 µL Blocker™ casein in PBS for 1 h on a shaking incubator at RT. The plates were then washed six times with 200 µL of 0.05% PBST and tapped dry. All subsequent wash steps follow the same procedure. Serum and positive controls were added in duplicate at a 1:50 dilution in 50 µL Blocker™ casein in PBS and incubated for 1 hr, shaking at RT (each plate contained a human and goat antiserum control for intra-assay validation). Plates were washed and tapped dry.

The detection/secondary antibody (either anti-human or anti-goat, Supplemental Table 8) was then added to the plate at 1:2000 in 100 µL, as recommended by the suppliers, and incubated for 30 min at RT. The plates were washed and tapped dry, and 100 µL of BioFX® TMB One Component HRP Microwell Substrate was added. The plates were allowed to develop for 15 min before stopping the reaction with 100 µL of BioFX® 450 nm Stop Reagent for TMB Microwell Substrates to prevent over-saturation. Absorbance was read at an optical density of 450 nm with a GloMax® Explorer Multimode Microplate Reader. Results were analysed manually (background subtraction and averaging of replicates) and plotted in R Studio[46] with ggplot2[47], ggstatsplot[48], and R 4.2.0[49] and GraphPad Prism 10.0.3.

Polyclonal goat antiserum, post-H5 vaccination pooled human plasma, and H1N1 convalescent sera were used as positive controls (Supplemental Table 8). The Scottish blood samples were used as a control cohort which was presumed to be H5 negative. The urban Malaysian cohort was used to understand baseline binding to H5 HA in a more urban population from Borneo (Supplemental Fig. 5).

## Pseudotyped influenza virus production

Pseudotyped lentiviruses displaying IAV H5 haemagglutinins were produced via transfection of the human embryonic kidney (HEK) 293T cells (Sigma, 12022001), as described in Thompson et al. [50] and Temperton et al. [30]. 1.0 µg of gag/pol construct (p8.91), 1.5 µg of a luciferase reporter carrying construct (PCSFLW), and 1.0 µg of HA glycoprotein expressing construct were combined with 200 µL Opti-MEM (Gibco™, 31985062) and 35 µL of 1 mg/mL polyethyleneimine (PEI) branched (Sigma, 408727). The p8.91, pCSFLW and the HA-expressing plasmids were obtained from CPT and NT; the A/chicken/Malaysia (Sabah)/6123/2018 HA plasmid, which was obtained from Twist Biosciences (USA).

Transfections were performed in 10 mL of Dulbecco's modified Eagle medium (DMEM) 1X, high glucose, GlutaMAX™ (Gibco™, 10569010) with 10% foetal bovine serum (FBS) (Gibco™, 16000044) and 1% penicillin–streptomycin (Sigma-Aldrich, P4333) and left for 24 h. One unit of neuraminidase from *Clostridium perfringens* (Sigma-Aldrich, N2876) was added to fresh media to induce virus budding. After 48 h, pseudotyped viruses were removed and filtered with a sterile 0.45-µm Millex®-HA filter unit (Millipore, SLHA033SB). The HA glycoprotein constructs used for the production of the pseudotyped H5N1 lentiviruses were A/Indonesia/05/2005 (Clade 2.1), A/Bar headed goose/Qinghai/1A/2005 (Clade 2.2), A/chicken/Malaysia(Sabah)/6123/2018 (Clade 2.3.2.1c).

The pseudotyped influenza viruses were titrated for use in microneutralisation assays. Pseudotyped influenza viruses with initial titres above $1 \times 10^5$ relative light units (RLU) were used (Supplemental Fig. 9).

## Pseudotyped microneutralisation assay

Neutralisation of the pseudotyped influenza viruses was quantified using a microneutralisation assay. Sera was added in duplicate at a 1:20 dilution in 100 µL DMEM to a white Nunc™ MicroWell™ 96-well, flat-bottom microplate. Each serum sample was then serially diluted down the plate (Row A–Row H) 1:1 in 50 µL DMEM. The plates were incubated for 2 h at 37 °C with 50 µL $1 \times 10^5$ RLU pseudotyped influenza virus per well. One row per plate contained virus, cells, and no serum, and one row was a cell-only control.

After 2 h, 50 µL of $1 \times 10^5$ HEK 293T cells were added to each well and incubated for a further 48 h at 37 °C. The cells were then lysed with 50 µL per well of a 1:1 mixture of Bright-Glo™ Luciferase Assay System and 1X PBS. The RLU of the cell lysate was determined using a GloMax® Explorer Multimode Microplate Reader.

Microneutralisation assay data was analysed in GraphPad Prism 10.0.3. The reduction of infectivity from the microneutralisation assay was determined by comparing the RLU in the presence and absence of serum and expressed as percentage neutralisation. The data was normalised using the virus and cell-only control wells, and a neutralisation curve was fit with non-linear regression (log-inhibitor versus normalised response). The 50%-neutralising titre (NT50) is here defined as the sample dilution at which viral RLU was reduced by 50% compared with control wells.

A sample was considered neutralising if the dilution factor of serum needed to reach NT50 was greater than or equal to 1:100, as established by Temperton et al. [30]. To account for cross-reactive neutralisation we used the highest NT50 from the cohort of negative control Scottish blood donor samples to create cut-offs for seropositivity (1:1173 for A/Indonesia/05/2005, 1: 1015 for A/Bar headed goose/Qinghai/1A/2005, and 1:1358 for A/Chicken/Malaysia(Sabah)/6123/2018) as in Thompson et al. [31] (Supplemental Fig. 4). Where applicable, NT50s were compared in Prism using the Extra sum-of-squares $F$-test, with $p < 0.05$ as the cut-off for statistically significant differences in NT50 values.

Two post-H5 vaccination pooled human plasma samples were used as positive controls. The two H1N1 convalescent samples were used as a control that exhibited a cross-reactive neutralising response (Supplemental Table 8). The Scottish blood samples were used as a control cohort which was presumed to be H5 negative (Supplemental Fig. 3).

## Microsphere (bead) protein conjugation

Carboxyl magnetic particles (beads) (Spherotech, SPHERO™ 4.2 µm) were conjugated with IAV HAs as described in Brown et al. [51], Barrett et al. [52], and Tomic et al. [53] for plasma depletion assays. Beads were coated with HAs from a variety of seasonal influenzas obtained from BEI Resources: A/Wisconsin/67/2005(H3N2), A/California/07/2009(H1N1) and A/New York/18/2009(H1N1). A subset of beads prepared were not coated with any HA to test the effect of the beads themselves in assays (referred to as non-coated beads).

All previously prepared beads were diluted to 200 beads/µL in an Assay Buffer (buffers described in Supplemental Table 9). 50 µL of each solution of beads was added to a 5 mL sterile, round bottom, polystyrene Falcon® test tube (Corning, 352054). Four conditions were prepared to test each bead: a blank tube with no secondary antibody, a positive control serum tube, a negative control serum tube, and an IgG isotype tube to identify any non-specific binding of the secondary antibody. The positive control serum was an individual who also

exhibited high H1 and H3 antibody binding responses by ELISA relative to the negative control. The negative control serum exhibited lower antibody binding to the seasonal HA antigens by ELISA. Responses by ELISA were used to establish controls, as past influenza subtype exposure is often unknown.

Serum samples were diluted 1:100 (50 μL) in Assay Buffer and incubated with beads for 1 h at RT, shaking at 700 rpm. Beads were pelleted on the EasyEights™ EasySep™ magnet (STEMCELL™ Technologies, 18103) for 1 min and the supernatant was gently aspirated without disturbing the bead pellet. Beads were washed three times using 1 mL Wash Buffer per tube. Beads were resuspended in 200 μL Assay Buffer. 100 μL of 1 μg/mL phycoerythrin (PE)-conjugated secondary antibody was added to the tubes, either the isotype control antibody (BioLegend #400112) or anti-human IgG (Southern Biotech #9040-09) depending on the condition. No secondary antibody was added to the blank condition. All tubes were then left to incubate for 1 h at RT in the dark, shaking. Beads were pelleted on a magnet and washed three times in 1 mL Wash Buffer. Samples were resuspended in 500 μL of Storage Buffer and vortexed.

Samples were then acquired on the flow cytometer to determine if the conjugation of viral proteins to the beads was successful. Side scatter versus forward scatter plots were used to draw gate 1 to exclude debris and doublets. The count versus PE histogram was used to observe PE fluorescence in the different conditions (Supplemental Fig. 4). Data was analysed with FlowJo™ 10.8.2.

### Bead-based cross-reactivity depletion

To determine the cross-reactivity of any H5 IAV neutralising and binding antibodies, a serum depletion assay was developed using the previously HA-conjugated beads. The aim was to isolate antibodies that bound to seasonal IAV-coated beads and determine if any H5 binding or neutralising ability remained.

For pseudotyped neutralisation assays, the bead mixture was diluted to a concentration of 100 beads/μL/bead (total concentration of 300 beads/μL). An equal concentration of A/Wisconsin/67/2005(H3N2), A/California/07/2009(H1N1), and A/New York/18/2009(H1N1) coated beads were used. Human serum was added to this bead mixture at a 1:20 dilution. This was incubated for 30 min, and the mixture was incubated on the magnet. The flow-through (media and serum not bound to the magnet) was transferred to an Eppendorf™ tube with fresh beads. This process was repeated four more times, for a total of five passages through fresh beads. Five passages generated the optimal depletion (highest level of depletion whilst conserving resources) (Supplemental Fig. 10a).

After the final passage, the mixture was incubated on the magnet for 1 min. While on the magnet, 100 μL of flow-through was added in duplicate to Row A of a white Nunc™ MicroWell™ 96-well plate and serially diluted to Row H. Each plate also contained untreated serum, a virus-only control row, and a cell-only control. The neutralisation assays were then completed and analysed as per the previous neutralisation results. NT50s of the treated and untreated conditions were compared in Prism using the Extra sum-of-squares F-test, with $p < 0.05$ as the cut-off for statistically significant differences in NT50 values.

The effect of the beads themselves was also tested at the final assay concentration (300 beads/μL) using non-coated beads. No effect was observed with the uncoated beads (Supplemental Fig. 10b).

As in the neutralisation assays, two post-H5 vaccination pooled human plasma samples were used as positive controls. The two H1N1 convalescent samples were used as a control that exhibited a cross-reactive neutralising response (Supplemental Table 8).

### Environmental risk factor analysis

We aimed to assess the spatial distribution of H5 binding, wild shorebird populations, and domesticated poultry. Binomial generalised linear models (GLMs) were used with the outcome as the proportion of

individuals per household with high H5 binding. For poultry ownership the number of poultry owners per village was used as an outcome. For wild shorebird contact, the outcome was whether there was a shorebird sighting at a particular location.

Households included in the study were geolocated and integrated with remote sensing-derived environmental data on land cover and climatic factors. All environmental data was curated as in Klim et al. [27] and Fornace et al. [43,54]. Twenty environmental variables were considered for inclusion in the final models of shorebird contact and H5 binding. Elevation, aspect, and slope data for the region were obtained from the ASTER Digital Global Elevation Model[55]. The WorldClim[56] database was used to collect data from 1970 to 2000 on average temperature, minimum temperature of coldest month, maximum temperature of warmest month, mean diurnal range (all in °C), precipitation seasonality (coefficient of variation), precipitation of wettest month (mm), and population density (per km²). Household distance in meters from mangroves, agricultural land, irrigated farmland, the sea, old (primary) forest, bush (secondary) forest, oil palm plantations, rubber plantations, and Euclidean distance from roads was calculated by Fornace et al. [43]. The possible predictor variables were mean-centred and scaled and checked for collinearity before inclusion in final models (Supplemental Fig. 8).

Fixed effects were selected by using the model with the lowest AIC was selected using MASS::stepAIC using a stepwise parsimonious approach (both forward and backward selection)[57] (Supplemental Tables 2–4). Odds ratios were calculated and plotted using the sjPlot[58] package in RStudio[46,49].

Spatial autocorrelation of the residuals from the GLM results was determined with Moran's I, where $p < 0.05$ was considered statistically significant. Models exhibiting residual spatial autocorrelation were integrated into a Bayesian framework with integrated nested Laplace approximations (INLA) in R-INLA[59]. Spatial effects were modelled as a Matérn covariance function, using the stochastic partial differential equation (SPDE) method[60]. For the model intercepts and fixed effects coefficients, weakly informative priors of Normal (0, 100) were used[61].

The final models were evaluated using the deviance information criteria (DIC). Posterior probabilities were estimated using 1000 posterior samples. These posterior probabilities were then used to predict the probability of the outcome variable (H5 binding or species distribution) across the whole study region. Uncertainty for these predictions was visualised through standard deviation. Posterior probabilities at 30 m spatial resolution (i.e. each pixel represents 30 m by 30 m) were visualised with ggplot2[47] and the wesanderson colour palette (copyright Karthik Ram, 2022).

### Reporting summary

Further information on research design is available in the Nature Portfolio Reporting Summary linked to this article.

## Data availability

The environmental data generated in this study have been deposited on HK's GitHub under accession code [https://doi.org/10.5281/zenodo.13767296]. The survey and serological data are available under restricted access. As the data includes identifiable information and household coordinates, data can be obtained with approval from relevant ethics committees in Malaysia and the UK. Please contact the corresponding authors for further details. Source data are provided as a Source Data file. Source data are provided with this paper.

## Code availability

Sample code and input data for the analyses in this study are available on HK's GitHub [https://github.com/hklim06/Serological-analysis-in-humans-in-Malaysian-Borneo-suggests-prior-exposure-to-H5-avian-influenza].

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

## Acknowledgements
We would like to thank the Director General of Health, Malaysia for permission to publish this study. We acknowledge the UK Medical Research Council, Natural Environment Research Council, Economic and Social Research Council and Biotechnology and Bioscience Research Council for funding received through the Environmental and Social Ecology of Human Infectious Diseases Initiative (Grant No. G1100796). KMF is supported by a Sir Henry Dale fellowship jointly funded by the Wellcome Trust and Royal Society (Grant No. 221963/Z/20/Z). This research was supported in part by US Food and Drug Administration Medical Countermeasures Initiative (contract 75F40120C00085), and the UK Public Health Rapid Support Team (Grant No. PSR00130). We acknowledge the reagents obtained through BEI Resources, NIAID, NIH and the specific providers listed in Supplemental Table 6. HK is supported by the Future of Humanity Institute at the University of Oxford DPhil Scholarship programme.

## Author contributions
H.K. undertook serological assays, geostatistical analysis, and study design. T.H.C. and T.W. collected data and contributed to the study design. J.M. assisted with experimental design and MNA assays. C.B. assisted with experimental design and flow cytometry experiments. G.S.R. contributed to conceptualising the study. H.B.M. assisted with serological assays. T.T. assisted with experimental design and serological assays. J.L.J., M.S.J., and K.A. provided Kota Kinabalu control samples and facility use at UMS BMHRC, and K.A. contributed to the study design. C.P.T., N.T., and K.D. assisted with the experimental design and contributed reagents. K.M.F., C.J.D., and M.W.C. conceptualised the study and analysed the data. K.M.F. performed geostatistical analyses.

## Competing interests
The authors declare no competing interests.

## Additional information

¹Nuffield Department of Medicine, Centre for Human Genetics and Pandemic Sciences Institute, University of Oxford, Oxford, UK. ²Infectious Diseases Society Sabah-Menzies School of Health Research Clinical Research Unit, Kota Kinabalu, Malaysia. ³Gleneagles Hospital, Kota Kinabalu, Malaysia. ⁴Clinical Research Centre, Queen Elizabeth II Hospital, Kota Kinabalu, Malaysia. ⁵Faculty of Medicine and Health Sciences, University of Malaysia Sabah, Kota Kinabalu, Malaysia.

[6]EduLife Berhad, Penampang, Sabah, Malaysia. [7]Faculty of Infectious and Tropical Diseases, London School of Hygiene and Tropical Medicine, London, UK. [8]Borneo Medical and Health Research Centre, Faculty of Medicine and Health Sciences, University of Malaysia Sabah, Kota Kinabalu, Malaysia. [9]Viral Pseudotype Unit, Medway School of Pharmacy, Universities of Kent and Medway, Kent, UK. [10]Department of Public Health Medicine, Faculty of Medicine and Health Sciences, University of Malaysia Sabah, Kota Kinabalu, Malaysia. [11]Division of Biomedical Sciences, Warwick Medical School, University of Warwick, Coventry, UK. [12]Department of Pathology and Microbiology, Faculty of Medicine and Health Sciences, University of Malaysia Sabah, Kota Kinabalu, Malaysia. [13]Research Center for Global and Local Infectious Disease, Oita University, Oita, Japan. [14]Saw Swee Hock School of Public Health, National University of Singapore, Singapore, Singapore. ✉e-mail: hannah.klim@ndm.ox.ac.uk; kfornace@nus.edu.sg

