## [Peer Review File · Nature Communications]

Reviewers' Comments:

Reviewer #1:

Remarks to the Author:

The study describes a serological survey in Sabah, Malaysian Borneo, to investigate human influenza exposure, mainly focusing on the immunological effects of H5N1 in the area. A method was introduced to mitigate the impact of influenza subtype cross-reactivity on the serological findings. The research defined the distributions of domesticated poultry and migratory wild birds, using environmental factors as proxies to model wild-bird contact. The study aims to highlight the surveillance of rare zoonotic diseases at migratory sites and proposes an approach for modeling the distributions of serological results and reservoir species. The study is interesting and relevant to the field, suggesting influenza exposure to individuals living near bird migratory sites.

However, the authors should address some points to improve the manuscript.

The introduction states the 2020-2024 epidemic wave caused by 2.3.4.4b H5N1 viruses worldwide. Despite this, the only recorded outbreak of was caused by a clade 2.3.2.1c HPAI H5N1 virus in a commercial poultry farm in Sabah state in 2018. It needs to be clarified why the authors used serum collected in 2015 and antigens of clade 2.1, 2.2, 2.3.4, and 2.3.4.4 if the outbreak in the Malaysian territory was caused by another clade, especially if the hypothesis was to investigate human influenza exposure. A review of the H5 detection in Malaysia and neighboring countries can help in the author's statement.

It would also be also interesting if the authors address whether poultry workers are vaccinated against seasonal influenza viruses.

Figure 2A. Although the authors state there is a significant difference between Individuals living in proximity to wild birds, it is not clear that the titers were higher for the individuals living in proximity to wild birds using the A/snow goose/Missouri/CC15-84A/2015(H5N2) virus.

Figure 2B: The authors state the work's limitation is that it does not have the same antigens for ELISA and microneutralization assay. Could you please add in the M&M section a brief sentence explaining this and add the respective clades of A/goose/Qinghai/1A/2005(H5N1) and A/Indonesia/05/2005(H5N1) antigens in the M&M text and the supplemental Table 1?

Please use the same scale of NT50 and absorbance at OD450 when comparing antigens in Figure 2.

Line 119: approximately 7% heterogeneity. Did you mean 93% of identity? Or 7% of distance based on amino acid sequences? Please revise the text.

Supplemental Table 1. This study used a phylogenetic analysis of hemagglutinins. Why did the authors not compare with viruses from clade 2.3.2.1c that caused an outbreak in 2018?

Supplemental Figure 1. Correlation matrix of ELISA binding results. Is it surprising that

the ELISA binding values are higher when comparing H3 and H5 to H1 and H5 because H1 and H5 hemagglutinin belong to the same genetic group 1? Please address this comment in the manuscript.

Supplemental Figure 2. Creation of seropositive cut-offs for pseudotyped neutralization assays. Please use the same scale for the figures in axis Y.

Reviewer #2:

Remarks to the Author:

The manuscript “Serological analysis in humans in Malaysian Borneo suggests prior exposure to H5 avian influenza” presents data from a cross sectional serum sample set. While similar studies have been conducted elsewhere, this study does utilize different serologic assays (albeit somewhat piecemeal) and potential exposure data. Specific comments follow.

1) It is somewhat unclear to me how “recent deforestation may increase the proximity between humans and wild birds”. Esp as the birds of focus in this work seem to be shorebirds (see next point).

2) The authors mention both migratory birds and migratory shorebirds. The distinction is important (obviously shorebirds are a subset of birds) and shorebirds have not really been implicated in the long range spread of H5. Do shorebirds and migratory waterfowl, far more common hosts of H5 viruses, share similar habitat in the study area. Could detection of shorebirds simply be a proxy for presence of waterfowl?

3) Ln 45. Many consider there is evidence for sustained community transmission of H5 within various sea mammal populations.

4) Ln 141. Were the authors not somewhat surprised that there was no correlation between the two H5 antigens? Especially in an ELISA.

5) Figure 2a. It would be useful to maybe use a method to display the data in panel A where the average can be seen. As an example of why, the authors state that the reactivity to the snow goose was higher in those in proximity to wild birds. The figure looks like it is actually the other way around.

6) Figure 2b. The cutoff line for positivity should be included here.

7) Overall I found it a bit difficult to determine exactly what serum panels were used in which assay. This could be better defined in the methods. What were the ELISA values for positive and negative control samples?

8) The values for seasonal viruses pre and post pulldown should be provided for the H5 positive samples to show pull down has worked (not just for the high titer positive samples).

Reviewer #3:

Remarks to the Author:

In this manuscript, Klim et al integrated serological, shorebird sighting, and environmental data to investigate factors associated with human H5 avian influenza infection. The spatial and environmental overlap of individuals with migratory birds increases the risk of H5 binding, offering novel insights into preventing cross-species transmission of avian influenza pathogens. Here are my comments and hope these can help improve the current manuscript.

Major:

1. Most reported cases of human infection with avian influenza are transmitted from poultry. Interestingly, this study found spatial and environmental overlap between individuals with high H5 binding and the distribution of migratory birds, the poultry and high H5 binding showed no significant relationship. This conclusion requires further explanation and discussion.

2. If H5 seroprevalence is associated with wild bird, one would expect variation before and after the migration of wild birds. Is this pattern evident in the serological survey data? For example, an increase in H5 seropositive samples may be observed following the arrival of wild birds. However, concentrated sampling in an area post wild bird arrival may introduce sampling bias. Therefore, the authors are encouraged to provide a comprehensive description of the serological survey data, including the temporal distribution and age structure of samples, and to account for any potential sampling bias.

3. Is there a linear relationship among the environmental predictor variables? What is the source of these data? It is important to consider poultry farming factors, such as poultry density, in this analysis.

Minor:

1. Line 61-62: It would be beneficial to provide quantitative data regarding the environmental changes in the region over the past 50 years. For instance, stating that the forest area has decreased by 20 percent would offer a clearer understanding.

2. Line 97-100: Given the inclusion of 500 samples from each of the four regions, it is essential to assess whether the population structures across these regions are comparable. Given that avian influenza cases commonly impact older adults and children, demographic variations could potentially influence prevalence rates and should be thoroughly considered and addressed.

3. Line 102-103: Is there overlap between individuals residing within a 10km radius of migratory shorebird habitats and poultry owners? How does the author group poultry farmers who live in close proximity to migratory birds?

4. Line 113-117: Why did you choose these two variants? Have they been observed circulating among poultry or wild bird populations in the study area?

5. Line 224: Can you specify the location of Kudat? It would be helpful to mark it on the map. Also, what does "n" mean? Number of samples? Please explain here.

6. Line 404: Which geographical area does Figure 4a show? Is there a spatial relationship between Figure 4a and Figure 1? If so, please elucidate this connection.

We thank the reviewers for their detailed comments on the manuscript. We have addressed their points below. Please note that line numbers correspond to the Word Document article file with tracked changes.

Reviewer #1 (Remarks to the Author):

The study describes a serological survey in Sabah, Malaysian Borneo, to investigate human influenza exposure, mainly focusing on the immunological effects of H5N1 in the area. A method was introduced to mitigate the impact of influenza subtype cross-reactivity on the serological findings. The research defined the distributions of domesticated poultry and migratory wild birds, using environmental factors as proxies to model wild-bird contact. The study aims to highlight the surveillance of rare zoonotic diseases at migratory sites and proposes an approach for modeling the distributions of serological results and reservoir species. The study is interesting and relevant to the field, suggesting influenza exposure to individuals living near bird migratory sites. However, the authors should address some points to improve the manuscript.

The introduction states the 2020-2024 epidemic wave caused by 2.3.4.4b H5N1 viruses worldwide. Despite this, the only recorded outbreak of was caused by a clade 2.3.2.1c HPAI H5N1 virus in a commercial poultry farm in Sabah state in 2018. It needs to be clarified why the authors used serum collected in 2015 and antigens of clade 2.1, 2.2, 2.3.4, and 2.3.4.4 if the outbreak in the Malaysian territory was caused by another clade, especially if the hypothesis was to investigate human influenza exposure. A review of the H5 detection in Malaysia and neighboring countries can help in the author's statement.

It would also be also interesting if the authors address whether poultry workers are vaccinated against seasonal influenza viruses.

We thank the reviewer for their comments and agree it would be of interest to look at serum from a later timepoint, however, the serum used in this study was collected in 2015 as a part of a different study (referenced on line 570). In response to the reviewer comments, we have now included 2.3.2.1c pseudovirus as a part of our study in pseudotyped neutralization assays (Figure 2b). On lines 202-213 we present the results of these new experiments, which had a 6.7% seropositivity rate (6.8% seropositivity was observed against A/Bar headed goose/Qinghai/1A/2005 pseudovirus, and 1.5% seropositivity was seen against A/Indonesia/05/2005 pseudovirus). We discuss that these results are in line with our expectations based on phylogenetic distance (lines 392-406).

Influenza vaccination rates in this region are low, as influenza vaccines are not a part of Malaysia's vaccination program. We have now added information in the text to reflect this (lines 331-340).

Figure 2A. Although the authors state there is a significant difference between Individuals living in proximity to wild birds, it is not clear that the titers were higher for the individuals living in proximity to wild birds using the A/snow goose/Missouri/CC15-84A/2015(H5N2) virus.

Apologies, we have now adjusted the text to clarify and avoid confusion (lines 132, 136, 137, 143, 144, 199). The results reflected in the text are medians and the comparisons are of mean ranks. The mean rank is compared in the Mann-Whitney test. For non-parametric data, the median is the appropriate measure of central tendency to report, as it better captures the distribution of the data (this is what was originally report in the text, but it was mislabeled as the mean). To assist in visualizing the medians, we have added black dashed lines to visually show these on the figures. We have also included the median values in Figure 2.

Figure 2B: The authors state the work's limitation is that it does not have the same antigens for ELISA and microneutralization assay. Could you please add in the M&M section a brief sentence explaining this and add the respective clades of A/goose/Qinghai/1A/2005(H5N1) and A/Indonesia/05/2005(H5N1) antigens in the M&M text and the supplemental Table 1?

We have now added several statements in the M&M clarifying the clades of each HA in the study (lines 611-612, 656-658). We originally selected recombinant protein and plasmids based on availability of materials and year of circulation. We chose one HA for ELISAs that was before and after the time of sample collection. Based on the reviewer's comments, we have now included the virus responsible for the 2018 poultry outbreak: A/chicken/Malaysia(Sabah)/6123/2018(H5N1) for the pseudotyped neutralisation and cross-reactivity depletion assays (Figure 2b).

Please use the same scale of NT50 and absorbance at OD450 when comparing antigens in Figure 2.

We have now adjusted the scale on the figure legends.

Line 119: approximately 7% heterogeneity. Did you mean 93% of identity? Or 7% of distance based on amino acid sequences? Please revise the text.

We have now adjusted the text to reflect this (line 128-129).

Supplemental Table 1. This study used a phylogenetic analysis of hemagglutinins. Why did the authors not compare with viruses from clade 2.3.2.1c that caused an outbreak in 2018?

The outbreak of the 2.3.2.1c was in the Tuaran district of Sabah, approximately 50km outside of our study site. Based on the reviewers' comments, we have now included pseudotyped neutralization assays presenting this 2.3.2.1c hemagglutinin in our study (Figure 2b). On lines 202-213 we present the results of these new experiments, which had a 6.7% seropositivity rate (6.8% seropositivity was observed against A/Bar headed goose/Qinghai/1A/2005 pseudovirus, and 1.5% seropositivity was seen against A/Indonesia/05/2005 pseudovirus). We discuss that these results are in line with our expectations based on phylogenetic distance on lines 392-406.

Supplemental Figure 1. Correlation matrix of ELISA binding results. Is it surprising that the ELISA binding values are higher when comparing H3 and H5 to H1 and H5 because H1 and H5 hemagglutinin belong to the same genetic group 1? Please address this comment in the manuscript.

We were also surprised by this result, especially as the correlations were different in the Malaysian versus Scottish study cohorts (we have now included this as Supplemental Figure 2b). We have expanded on this in the text (lines 152-158, 349-364).

Supplemental Figure 2. Creation of seropositive cut-offs for pseudotyped neutralization assays. Please use the same scale for the figures in axis Y.

We have now adjusted the scaled for this figure (now Supplemental Figure 3).

Reviewer #2 (Remarks to the Author):

The manuscript “Serological analysis in humans in Malaysian Borneo suggests prior exposure to H5 avian influenza” presents data from a cross sectional serum sample set. While similar studies have been conducted elsewhere, this study does utilize different serologic assays (albeit somewhat piecemeal) and potential exposure data. Specific comments follow.

1) It is somewhat unclear to me how “recent deforestation may increase the proximity between humans and wild birds”. Esp as the birds of focus in this work seem to be shorebirds (see next point).

Deforestation in this region includes destruction of mangrove and swamp forests, which are habitats for migratory shorebirds and waterfowl (originally described on lines 63-67). We have now clarified in the text that the deforestation relevant to this study is coastal deforestation (lines 9, 35). In this study, we looked at proximity to different habitats which may be impacted by deforestation.

2) The authors mention both migratory birds and migratory shorebirds. The distinction is important (obviously shorebirds are a subset of birds) and shorebirds have not really been implicated in the long range spread of H5. Do shorebirds and migratory waterfowl, far more common hosts of H5 viruses, share similar habitat in the study area. Could detection of shorebirds simply be a proxy for presence of waterfowl?

We have now clarified in the text that we are referring to migratory shorebirds. There is evidence suggesting that migratory shorebirds have been implicated in the long range spread of H5, particularly in the East Asian-Australasian flyway (the migratory path that includes the island of Borneo), which we have now included on lines 458-477.

We agree with Reviewer 2 that waterfowl are likely to be found in similar habitats to migratory shorebirds. We have now added a map of waterfowl and shorebird locations in the region Supplemental Figure 7. Based on the data reported to eBird, there have been no sightings of waterfowl recorded in the northern coastal areas of Sabah. Although this does not mean there are no waterfowl in these areas, Supplemental Figure 7 supports the hypothesis that the presence of migratory shorebirds may be driving this spillover.

3) Ln 45. Many consider there is evidence for sustained community transmission of H5 within various sea mammal populations.

We have now amended this statement to reflect this (line 45-46).

4) Ln 141. Were the authors not somewhat surprised that there was no correlation between the two H5 antigens? Especially in an ELISA.

As noted in response to Reviewer 1, we were surprised by this result, especially as the correlations were different in the Malaysian versus Scottish study cohorts (we have now included this as Supplemental Figure 2b). In the Malaysian cohort, the two H5 antigens exhibit a weakly positive correlation ($r=0.22$). However, in the Scottish cohort, this correlation is close to zero ($r=0.02$). In the Scottish cohort, correlations between the H1 and H5 antigens are higher than in the Malaysian cohort. This is an interesting result, which we have now noted in the text (lines 152-158).

One possible explanation for this population level difference could be due to the difference in seasonal vaccinations or the seasonality of the virus itself, which does not experience monthly swings in tropical climates. The weak positive correlations between the H5 ELISA antigens could be due to the plate format of the ELISA exposing different epitopes.

5) Figure 2a. It would be useful to maybe use a method to display the data in panel A where the average can be seen. As an example of why, the authors state that the reactivity to the snow goose was higher in those in proximity to wild birds. The figure looks like it is actually the other way around.

We thank the reviewer for their comments, we have now adjusted the text to clarify and avoid confusion (lines 132, 136, 137, 143, 144, 199). The results reflected in the text are medians and the comparisons are of mean ranks. The mean rank is compared in the Mann-Whitney test. For non-parametric data, the median is the appropriate measure of central tendency to report, as it better captures the distribution of the data (this is what was originally report in the text, but it was mislabeled as the mean). To assist in visualizing the medians, we have added black dashed lines to visually show these on the figures. We have also included the median values in Figure 2.

6) Figure 2b. The cutoff line for positivity should be included here.
We have now adjusted Figure 2 to include these cutoff positivity lines.

7) Overall I found it a bit difficult to determine exactly what serum panels were used in which assay. This could be better defined in the methods. What were the ELISA values for positive and negative control samples?

We apologize for the confusion; this information was originally included in Supplemental Figure 4 (now supplemental Figure 5) and Supplemental Table 8. To make this clearer, we have now included a flow chart detailing our experimental methods and which samples were used for our experiments as Supplemental Figure 1. We have additionally defined the positive and negative controls for each assay in the methods (lines 633-636, 694-698, 778-780).

8) The values for seasonal viruses pre and post pulldown should be provided for the H5

positive samples to show pull down has worked (not just for the high titer positive samples).

We did not produce seasonal pseudotyped viruses for this study, but we did test pre and post depletion values using the seasonal ELISA antigens. We have now included this as Supplemental Figure 10c.

Reviewer #3 (Remarks to the Author):

In this manuscript, Klim et al integrated serological, shorebird sighting, and environmental data to investigate factors associated with human H5 avian influenza infection. The spatial and environmental overlap of individuals with migratory birds increases the risk of H5 binding, offering novel insights into preventing cross-species transmission of avian influenza pathogens. Here are my comments and hope these can help improve the current manuscript.

Major:

1. Most reported cases of human infection with avian influenza are transmitted from poultry. Interestingly, this study found spatial and environmental overlap between individuals with high H5 binding and the distribution of migratory birds, the poultry and high H5 binding showed no significant relationship. This conclusion requires further explanation and discussion.

We have now included Supplemental Table 7, which shows the inclusion of swine and poultry ownership as fixed effects did not improve model fit in our models of H5N1 HA binding. We additionally broadened our discussion of these findings and of the possibility of spillover from migratory shorebirds on lines 458-477.

2. If H5 seroprevalence is associated with wild bird, one would expect variation before and after the migration of wild birds. Is this pattern evident in the serological survey data? For example, an increase in H5 seropositive samples may be observed following the arrival of wild birds. However, concentrated sampling in an area post wild bird arrival may introduce sampling bias. Therefore, the authors are encouraged to provide a comprehensive description of the serological survey data, including the temporal distribution and age structure of samples, and to account for any potential sampling bias.

We thank the reviewer for their comments and agree that repeated sampling over the year would be informative. However, the samples available for this study were collected over a single time period, therefore, so we would not be able to compare before and after migration events with the data available to us. We agree that this would be interesting point for future study, as noted on lines 455-456 of the manuscript.

We have added more information on the ages of our sample cohort on lines 105-107 and explained our sample selection strategy and approach in Supplemental Figure 1. We have also now included the dates of sample collection on lines 570-571. This survey was designed to be representative of the entire population, including all age groups (see Fornace et. al, 2018 for more details).

3. Is there a linear relationship among the environmental predictor variables? What is the source of these data? It is important to consider poultry farming factors, such as poultry density, in this analysis.

We did not originally include this as our two previous papers referenced in this

section on line 792 detailed and presented this method. We have now provided more detailed information on the sources of these data (lines 792-805) and provided data on co-linearity in Supplemental Figure 8.

The density of poultry was considered as the distribution of poultry ownership in each village in the study site in Supplemental Figure 6.

Minor:

1. Line 61-62: It would be beneficial to provide quantitative data regarding the environmental changes in the region over the past 50 years. For instance, stating that the forest area has decreased by 20 percent would offer a clearer understanding.

We have now added a statement clarifying this on lines 63-64.

2. Line 97-100: Given the inclusion of 500 samples from each of the four regions, it is essential to assess whether the population structures across these regions are comparable. Given that avian influenza cases commonly impact older adults and children, demographic variations could potentially influence prevalence rates and should be thoroughly considered and addressed.

We have now included a statement on the average age in each region on lines 105-107. We have also added further demographic information in Supplemental Figure 1.

3. Line 102-103: Is there overlap between individuals residing within a 10km radius of migratory shorebird habitats and poultry owners? How does the author group poultry farmers who live in close proximity to migratory birds?

47% of those living within a 10km radius of migratory shorebird sightings are also poultry owners (now stated on lines 111-114). The samples were sorted independently for the comparisons in Figure 2. This means that samples are grouped exclusively on poultry ownership or proximity to wild birds depending on which comparison is being made. On lines 192-200, we explain the number of neutralizing samples from individuals who are both poultry owners and living in proximity to migratory birds.

4. Line 113-117: Why did you choose these two variants? Have they been observed circulating among poultry or wild bird populations in the study area?

There has only been one recorded previous outbreak of H5 HPAI recorded in Malaysian Borneo, which was of the 2.3.2.1c clade (lines 70-76). We did not have access to recombinant protein available for HAs of this clade. However, we have now incorporated this into our pseudovirus neutralisation assay (Figure 2b). The two recombinant proteins used in ELISAs were therefore selected based on the proteins that were available to us for this study. We selected one H5 HA from before and after the date of sample collection. Both H5 HAs used for this study come from

clades that have circulated in wild bird populations and poultry globally (Clades 2.3.4 and 2.3.4.4).

5. Line 224: Can you specify the location of Kudat? It would be helpful to mark it on the map. Also, what does "n" mean? Number of samples? Please explain here.

We now have clarified this line in the text (line 272). We have also included a statement on this in the methods (lines 595-596). We will defer to the editor on what standard format to present this in. We have included a map of the four districts in Supplemental Figure 1.

6. Line 404: Which geographical area does Figure 4a show? Is there a spatial relationship between Figure 4a and Figure 1? If so, please elucidate this connection.

Figure 4a predicted estimated mean probabilities across the study site shown in Figure 1. We have now included this in the Figure 4 caption (line 549). We have also included a map defining the entire geographic area in Supplemental Figure 1.

Reviewer #1

(Remarks to the Author)

The authors performed additional experiments and analyses in the revised version to address the reviewers' comments. The text was also refined to emphasize the findings better, improving the manuscript. The study underscores the importance of investigating cross-species pathogen transmission in migratory zones and its relevance to the ongoing H5N1 epizootics. Therefore, the publication of the manuscript is recommended.

(Remarks to the Editor)

Sorry for the delay

Reviewer #2

(Remarks to the Author)

the authors have addressed my primary concerns.

(Remarks to the Editor)

Reviewer #3

(Remarks to the Author)

Thank you for addressing my comments and suggestions. The improvements made have significantly enhanced the clarity and impact of your study.

I have just a few minor points for your consideration:

1. The authors suggest that contact with avian species is currently necessary for a spillover event. However, the analysis does not directly compare the impact of wild shorebird contact and poultry density on the probability of high H5 binding. It might be helpful to consider incorporating variables such as wild shorebird contact and poultry density to better assess the roles of these factors in virus transmission.

2. Could you please clarify the spatial resolution of Figure 4a and how the data were generated? Additionally, in Figure 4b, could you explain what the different colors of the points (red and blue) represent?

3. The authors investigated H5 HA binding in 2015, but the GLM analysis utilizes meteorological data from 1970-2000. Would it be possible to consider using

meteorological data that aligns more closely in time with the serological data?

4. To provide readers with a clearer understanding of the location, it may be helpful to mark the geographic range of Northern Sabah in Figure 1.

We thank the reviewers for their positive responses to our changes to the manuscript. We have included replies to Reviewer 3's comments below.

Reviewer #3 (Remarks to the Author):

Thank you for addressing my comments and suggestions. The improvements made have significantly enhanced the clarity and impact of your study.

I have just a few minor points for your consideration:

1. The authors suggest that contact with avian species is currently necessary for a spillover event. However, the analysis does not directly compare the impact of wild shorebird contact and poultry density on the probability of high H5 binding. It might be helpful to consider incorporating variables such as wild shorebird contact and poultry density to better assess the roles of these factors in virus transmission.

In the main text of this study, we chose to highlight the modeled distributions and environmental risk factors for shorebird sightings and H5 contact. This is because we do not have data on H5 binding or shorebird sightings across every GPS coordinate, whereas we do have environmental data for each coordinate. This allows us to make predictions based on environmental covariates across the entire region as in Klim et al. 2022 (<https://doi.org/10.3389/fepid.2022.1057047>) and Chan et al. 2022 (<https://doi.org/10.3389/fpubh.2022.924316>).

In supplemental table 5, we incorporate the model for the wild shorebird distribution into the model of H5 binding with null, separate, and joint spatial effects. We have taken this approach to capture distributions and risk factors across the entire geographic region in our study. We demonstrate that a shared spatial effect between the distribution of shorebirds and H5 exposure improves model fit, suggesting a correlated processes.

We directly tested whether there was an association with poultry contact with increased exposure to H5. In supplemental table 7, we directly incorporate both poultry and swine ownership as fixed effects in the multivariate models and show that the inclusion of these covariates is not statistically significant.

2. Could you please clarify the spatial resolution of Figure 4a and how the data were generated? Additionally, in Figure 4b, could you explain what the different colors of the points (red and blue) represent?

We have now added more information in the figure caption on the spatial resolution of Figure 4a and how the data were generated to reflect the details in the methods section. We have now added a statement in the caption for Figure 4b explaining that odds ratios greater than 1 (shown in blue) indicate a factor that increases the odds of high H5 binding or of migratory shorebird sightings. Odds ratios less than 1 (shown in red) decrease the odds of those outcomes.

3. The authors investigated H5 HA binding in 2015, but the GLM analysis utilizes

meteorological data from 1970-2000. Would it be possible to consider using meteorological data that aligns more closely in time with the serological data?

As shorebird data was collected over multiple years and individuals may have been exposed at different time points, we used long-term averages from 1970-2000 data to capture climatic conditions rather than recent weather changes. These long-term datasets are widely used to for species distribution models as these represent the suitability of a habitat for a particular species over time. This also keeps our results consistent with our previously published studies (Klim et al. 2022, Chan et al. 2022, and Fornace et al. 2019).

4. To provide readers with a clearer understanding of the location, it may be helpful to mark the geographic range of Northern Sabah in Figure 1.

We have now added a statement in the caption of Figure 1 explaining that the sampled villages are all within the north of the Sabah region, which we hope will clarify this confusion.